# Profiling Genome-Wide DNA Methylation in Children with Autism Spectrum Disorder and in Children with Fragile X Syndrome

**DOI:** 10.3390/genes13101795

**Published:** 2022-10-04

**Authors:** Mittal Jasoliya, Jianlei Gu, Reem R. AlOlaby, Blythe Durbin-Johnson, Frederic Chedin, Flora Tassone

**Affiliations:** 1Biochemistry and Molecular Medicine, University of California, Sacramento, CA 95817, USA; 2School of Public Health: Biostatistics, Yale University, New Haven, CT 06510, USA; 3College of Health Sciences, California Northstate University, Sacramento, CA 95757, USA; 4Department of Public Health Sciences, Division of Biostatistics, University of California, Davis, CA 95616, USA; 5Davis, Genome Center, University of California, Davis, CA 95616, USA; 6Davis, Molecular & Cellular Biology, University of California, Davis, CA 95616, USA; 7MIND Institute, UC Davis, Sacramento, CA 95817, USA

**Keywords:** DNA methylation, fragile X syndrome, autism spectrum disorder, epigenetic, CpG

## Abstract

Autism spectrum disorder (ASD) is an early onset, developmental disorder whose genetic cause is heterogeneous and complex. In total, 70% of ASD cases are due to an unknown etiology. Among the monogenic causes of ASD, fragile X syndrome (FXS) accounts for 2–4% of ASD cases, and 60% of individuals with FXS present with ASD. Epigenetic changes, specifically DNA methylation, which modulates gene expression levels, play a significant role in the pathogenesis of both disorders. Thus, in this study, using the Human Methylation EPIC Bead Chip, we examined the global DNA methylation profiles of biological samples derived from 57 age-matched male participants (2–6 years old), including 23 subjects with ASD, 23 subjects with FXS with ASD (FXSA) and 11 typical developing (TD) children. After controlling for technical variation and white blood cell composition, using the conservatory threshold of the false discovery rate (FDR ≤ 0.05), in the three comparison groups, TD vs. AD, TD vs. FXSA and ASD vs. FXSA, we identified 156, 79 and 3100 differentially methylated sites (DMS), and 14, 13 and 263 differential methylation regions (DMRs). Interestingly, several genes differentially methylated among the three groups were among those listed in the SFARI Gene database, including the *PAK2*, *GTF2I* and *FOXP1* genes important for brain development. Further, enrichment analyses identified pathways involved in several functions, including synaptic plasticity. Our preliminary study identified a significant role of altered DNA methylation in the pathology of ASD and FXS, suggesting that the characterization of a DNA methylation signature may help to unravel the pathogenicity of FXS and ASD and may help the development of an improved diagnostic classification of children with ASD and FXSA. In addition, it may pave the way for developing therapeutic interventions that could reverse the altered methylome profile in children with neurodevelopmental disorders.

## 1. Introduction

According to the recently revised DSM-5, ASD is defined as a neurological and a neurodevelopmental disorder with deficits in two core symptom domains: (i) language, social communication and interaction, and (ii) restricted, repetitive behaviors [1]. Because of the genetic heterogeneity paired with a variable clinical presentation, ASD, with a prevalence of 1 in 59 children, is classified as an extremely complex disorder [2]. In ASD, the male-to-female ratio is 4:1 and the suggested protective/male susceptibility is supported by the observed excessive burden of de novo and inherited single nucleotide variants (SNVs), copy-number variants (CNVs) [3,4], and biallelic mutations, in females [5]. Approximately 75% of ASD cases have no known cause [6]. Genetic causes of ASD include de novo point mutations [7,8], common variants [9], rare de novo variants, copy number variations [10,11], recessive mutations [12], biallelic-loss of-function and missense mutations [5], particularly in genes involved in neurodevelopment and postzygotic mosaic mutations [13]. However, a great proportion of ASD cases is not explained by the above mechanisms; therefore, more studies are needed to assess the additional genetic contributions to ASD risk.

Importantly, ASD is thought to result from multiple interacting genes and epigenetic and environmental factors. Epigenetic modifications, with DNA methylation being the most well-characterized and studied, are pivotal to developmental and regenerative biology [14]. Several studies have suggested a key epigenetic role in the etiology of ASD, indicating that many genes linked to ASD encode for proteins that regulate chromatin remodeling, nucleosome assembly or DNA or histone modifications [15]. An array of ASD risk genes, including but not limited to, methyl CpG-binding protein 2 (MECP2), protein kinase C β gene (PRKCB1), oxytocin receptor gene (OXTR), B-cell lymphoma 2 (BCL2), retinoic acid-related orphan receptor α (RORA), SHANK3, β-catenin and zinc finger protein 57 (ZFP57) contribute to ASD pathogenesis when epigenetically altered [16,17,18]. These findings agree with several studies in which genes with a role in epigenetic pathways were found to represent a large percentage of the candidate genes involved in ASD [7,19,20].

Fragile X syndrome (FXS) is the most common monogenic cause of ASD with ~60% of FXS individuals presenting with autism spectrum disorder [21,22]. FXS is caused by a long (>200) CGG trinucleotide repeat expansion, which suppresses the *FMR1* gene transcription via hypermethylation of the promoter and of the repeat located within the 5′-UTR of the gene. This results in transcriptional shut-down and the absence or reduced expression of the encoded gene product, FMRP [23] mRNA-binding translational regulator and a modulator of synapse maturation and plasticity [24]. Further, multiple functions of FMRP have been validated, including editing, chromatin binding, microRNA, and gating of ion channels [25]. In addition, the involvement of FMRP in the pathogenesis of ASD has been demonstrated, which is consistent with the observed overlapping clinical manifestations of FXS and ASD. It has been reported that genes harboring ASD risk mutations are enriched in FMRP targets [26,27,28] and a lower FMRP expression has been detected in several neurodevelopmental disorders [29,30,31].

Several studies have demonstrated that alterations in epigenetic processes can lead to several neuropsychiatric conditions, including FXS and ASD [32,33]. In FXS, the epigenetic silencing of the fragile X messenger ribonucleoprotein 1 (*FMR1*) gene, specifically by DNA methylation and histone modifications, results in the loss of the encoded protein, FMRP, with genome-wide consequences due to the role of FMRP in regulating the expression of several coding and non-coding RNAs [34]. As a result, the regulation of transcription of many genes, including those playing a role in synaptic plasticity and neuronal functions, is affected in FXS.

There is a lack of studies showing differences in the DNA methylation profile of idiopathic ASD and FXS with ASD (FXSA). Thus, here we investigated the genome-wide DNA methylation profiles of children with FXSA, ASD, and compared them to TD children. Global methylation at 850,000 CpG sites spanning the whole genome was investigated to identify differentially methylated sites (DMS) and associated genes, potentially contributing to the pathogenesis of ASD and FXSA.

This is of importance, as it could lead to a better understanding of the pathways involved in these disorders, to independently confirming some of the already identified risk genes, to developing novel therapeutics approaches, and to the identification of potential biomarkers for the early detection of these conditions.

## 2. Materials and Methods

### 2.1. Study Participants and Samples

Biological samples were collected from 57 age-matched (2–6 years old) male participants, including 23 subjects with a diagnosis of ASD, 23 with a diagnosis of FXSA, and 11 TD children. The mean ages were 4.1 (±1.1 SD) and 3.6 (±1.1 SD) and 3.8 (±1.3 SD), respectively. Written informed consent was obtained from all individuals for the collection of biological samples under protocols approved by the UC Davis Institutional Review Board (IRB protocol number: 271070-4, 271070-29) following the ethical and legal regulations and principles of the Declaration of Helsinki. To confirm ASD diagnosis, both the ADOS [35] and the DSM-V checklist [36] were used. Demographic data, including age, CGG repeat number, and mutation category, are shown in Table 1.

### 2.2. CGG Allele Sizing and Methylation Status

Genomic DNA was extracted from 3 mL of peripheral blood by using the Gentra Puregene Blood Kit (Qiagen, Valencia, CA, USA). The CGG repeat allele size and methylation status were assessed using a combination of PCR and Southern blot analysis as previously described [37]. Briefly, for the Southern blot, 10 μg of gDNA was digested with EcoRI and NruI, run on an agarose gel, transferred on a nylon membrane, and hybridized with the *FMR1*-specific dig-labeled StB12.3. Southern blot analysis was also used to determine the methylation status of the *FMR1* alleles (percent of methylation) as previously described [38]. PCR, specifically target *FMR1* amplification (AmplideX PCR/CE, Asuragen, Austin, TX, USA), was used to determine the CGG repeat length; PCR amplicons were visualized by CE and analyzed as previously reported [39].

### 2.3. Data Processing

Data processing and analysis were performed in R (version 4.0.2) using minfi, limma, DMRcate, ChAMP, and methylCC package. As shown in Appendix A, we removed 30,435 (3.52%) probes containing a SNP or at the single nucleotide extension, and 723 probes that were not detected (detection *p*-value > 0.1) in more than 5% of the samples were excluded from the analyses. Additionally, we removed 8155 probes with multiple annotations as previously described [40]. After quality control, a total of 826,551 (95.46%) probes remained for analysis. All samples successfully matched as males when comparing the predicted sex with self-reported sex.

To eliminate between-array differences, a between-array (functional) normalization (implemented in pre-process Funnorm function of minfi package) was performed using control probes. After functional normalization, β values were calculated and used to represent DNA methylation levels, which is the ratio between the intensities of methylated versus unmethylated probes, ranging from 0 to 1. Next, we also performed a one-way ANOVA analysis for the PCs of β value matrix with disease status groups and possible known batch effect variables. Appendix A shows that the associations between PCs and array index, and row index of the array were significantly reduced. This could indicate that the functional normalization process improves the in-between-array differences.

### 2.4. Principal Component Analysis

To estimate possible batch effects, a principal component (PC) analysis was performed using the β value matrix. Appendix A shows that sex chromosomes are mainly attributed to the methylation profiles differences between FXSA vs. ASD samples and ASD vs. TD. After excluding the sex chromosomes, the samples from the three disease status groups were jumbled all together, and there were no overall differences in their autosomal methylation profiles. This could indicate that the disease status may be associated with DNA methylation variation at a few other specific sites in the genome.

### 2.5. Cell Type Composition Estimates

Each sample had estimated relative cell type proportions (CD8T, CD4T, natural killer cells, B cells, monocytes, and granulocytes cells) using a reference-free method implemented in the methylCC package [41]. As shown in Appendix A, neutrophil cells were the most abundant type of blood cells (on average 41.57%) in our samples. To assess whether cell composition was associated with the disease groups, we performed a Wilcoxon test, using the estimated cell proportions between the three diseases’, ASD, FXSA and TD, status groups. The FXSA samples showed a relatively higher proportion of CD4T and B cells, while the TD samples had relatively higher proportions of granulocytes. There were no significant differences observed for CD8T, monocytes and NK cells among the three disease groups.

### 2.6. Differentially Methylated Sites (DMS)

We performed differential methylation analysis with M values instead of β values, as the M values have reportedly shown better performance [42]. First, we conducted linear regression with M values using the R package limma as an unadjusted model. Second, we applied linear regression with M values and included several factors as model covariates:

M~Disease Status Group + Granulocytes + CD4T + BCell + Intercept

The *p*-values for regression coefficients in the disease status group variable, were subjected to Benjamini–Hochberg multiple-test correction, and the FDR values were obtained. Appendix A shows the QQ-plot indicating that, after adjusting for cell type composition, the genomic inflation for the *p*-values from an unadjusted model were significantly reduced. Moreover, the threshold of FDR <= 0.05 was used in the downstream analysis, resulting in 156, 79, and 3100 significant DMS for TD vs. ASD, TD vs. FXSA, and ASD vs. FXSA, respectively. Afterward, we filtered the probes based on the observed effect size. Two thresholds of β value between group-wise means were applied, 0.05 and 0.1, yielding DMS with the largest effects, respectively (Table 2).

### 2.7. Differential Methylated Regions (DMR)

Regions that were differentially methylated were characterized using M-values and R package DMRcate that first identified and combined DMS further calculates *p*-values (Stouffer’s method) with the Benjamini–Hochberg correction. The threshold of HMFDR ≤ 0.05 and the number of significant differential methylation sites within a genomic region ≥ 10 were used in the downstream analysis.

### 2.8. Gene Set Enrichment Analysis

We used 53, 60 and 545 significant DMS with an absolute effect size of β values at least 0.05 for gene set enrichment with the methylglm function from the methylGSA R package [43]. The FDR threshold was set to 0.001. We used REVIGO to summarize the GO terms [44].

### 2.9. Measurements of mRNA Expression Levels

Transcript expression levels for a subset of genes were measured by real time quantitative RT-PCR based on (a) their significant differential methylation in children with ASD and FXSA compared to TD, (b) their involvement in brain functions and the pathology of neurological disorders, and (c) their association with the promoter regions. Total RNA from whole blood was isolated using either Tempus tubes (Applied Biosystems, Waltham, MA, USA) or PAX gene tubes (Qiagen, Valencia, CA, USA) according to the manufacturer’s instructions. Total RNA quantification was performed using Nanodrop (Thermo Fisher Scientific, Waltham, MA, USA).

cDNA synthesis was performed in 100 μL aliquots containing 1 × PCR buffer (20 mM Tris-HCl, pH 8.4, 50 mM KCl) (Gibco/BRL), 5.5 mM MgCl_2_, 1 mM each dNTP, 5 μM random sequence deoxyoligonucleotide hexamers (Gibco/BRL), 0.4 U RNAse inhibitor (Gibco/BRL), and 2.5 U Moloney murine leukemia-virus RT (Gibco/BRL). At least three concentrations of total RNA (500 ng, 250 ng, and 125 ng per 100 μL reaction) were used for each sample, to ensure the linearity of the RT-PCR response. The RT temperature profile was as follows: 25 °C for 10 min, 48 °C for 40 min, 95 °C for 5 min, and final cooling to 4 °C. As a control for genomic contamination, 500 ng total RNA was treated as described above, with the exception that the RT was omitted.

Assays on demand for the selected genes (Applied Biosystems, Waltham, MA, USA) were used to measure their mRNA expression levels. Custom designed TaqMan primers and probe assays (Thermo Fisher Scientific, Waltham, MA, USA) were used to measure the expression levels of three internal control genes for normalization including β-glucuronidase (GUS), glyceraldehyde-3-fosfato dehydrogenase (GAPDH) and hypoxanthine phosphoribosyltransferase 1 (Hprt1).

### 2.10. Measurement of Protein Expression Levels

Plasma samples stored at −80 °C were thawed at room temperature and the expression levels of NF2 (Neurofibromin 2) and C11ORF31proteins were measured using human Neurofibromin/Merlin (NF2) ELISA Kit (MBS9331611) (MyBioSource, San Diego, CA, USA) and human Selenoprotein H (SelH) ELISA Kit (MBS9312777) (MyBioSource, San Diego, CA, USA respectively. Briefly, standard dilutions were made according to the manufacturer’s instruction. Samples and standards were incubated in a 96-well plate containing primary antibody for one hour at room temperature. This was followed by treatment with HRP conjugated secondary antibody and incubation with chromogen solutions whose absorbance was measured at 450 nm using the Biotek Synergy HT plate reader. The concentration of the unknown samples was measured using the slope of the linear standard curve.

## 3. Results

### 3.1. Subjects

A total of 57 samples collected from 11 TD, 23 ASD, and 23 FXSA male participants were included in this study. Fragile-X DNA testing ruled out the presence of an *FMR1* CGG expansion in both TD (*n* = 11) and ASD (*n* = 23) participants. Among the FXSA confirmed to have an *FMR1* allele with greater than 200 CGG repeats, 52.2% (*n* = 12) had a hypermethylated full mutation while 47.8% (*n* = 11) were mosaics for the presence of both methylated and unmethylated alleles (Table 1). Among the mosaics, 45.4% (*n* = 5) were methylation mosaics (methylated and unmethylated alleles spanning the entire expanded range), and 54.5% (*n* = 6) were size mosaics [with a premutation (55-200 CGG repeats) and a full mutation allele].

### 3.2. Differentially Methylated Sites (DMS)

The data analysis pipeline is shown in Appendix A. All samples were subjected to preprocessing and normalization aiming on minimizing technical variation and between-array differences. As a result, 39,313 (4.54%) probes were eliminated by various quality control criteria and 826,551 high-quality probes were retained for the downstream analysis. Functional normalization, as implemented in the minfi package [45], was used to remove between-array variance. After normalization, PCA analysis of the β values shows that the samples from the three disease status groups were jumbled all together, and there were no overall differences in autosomal methylation profiles of these three groups’ samples (Appendix A). This could indicate that disease status is associated with DNA methylation variation at few specific sites. Moreover, we estimated the cell type proportions of our whole blood samples and analyzed differences in the immune cell composition between the three disease status groups. Significant differences were identified in CD4T, B cell, and granulocytes. Correlation analysis confirmed variations in cell type compositions as a strong confounding factor (Appendix A). Finally, we included the cell type proportions in the linear regression model for the identification of DMS.

We performed differential methylation analysis with M values, as it is considered to have a better performance [42]. We assessed genomic inflation by the QQ-plot and observed that (Appendix A) after adjusting cell type proportions, the genomic inflations of the unadjusted model were effectively reduced. After applying a conservative discovery rate threshold (FDR ≤ 0.05), we identified 156 DMS between TD vs. ASD samples, 79 DMS between TD vs. FXSA samples, and 3100 DMS between ASD vs. FXSA samples (Table 2, Figure 1). To identify the DMS of greatest interest, we computed the absolute differences in mean methylation β-values between disease status groups, applying two effect size thresholds 0.05 and 0.1, and this identified a set of DMS with a large effect size (Table 2, Figure 2).

### 3.3. Differentially Methylated Regions (DMR)

To analyze the presence of neighborhoods of differential methylation, sites were tested for agglomeration of individual methylation sites into discrete, differentially methylated regions. In total, 14 DMRs (differentially methylated regions) were identified between TD vs. ASD samples, 13 DMRs between TD vs. FXSA samples, and 263 DMRs between ASD vs. FXSA samples. The regions included the *FMR1* gene, consistent with the methylation status of this locus. Among other regions identified as the most significantly differentially methylated among the groups, there were loci associated with homeobox genes, which play important roles in embryonic development and cell differentiation and have been implicated in human diseases and neurodegeneration [46,47].

### 3.4. Risk Genes for Autism Are Differentially Methylated and among Those Identified in the Three Groups

To determine if the genes associated with 53, 60, and 545 DMS (with high effect size, Table 2) identified in our analysis were associated with ASD, we checked if they were listed on the SFARI Gene database (which contains genes associated with ASD). Two genes, *PAK2* and *FANCD2* differentially expressed in TD vs. ASD, three genes, *DNMT3A*, *FOXP1*, and *GTF2I* differentially expressed in TD vs. FXSA, and seven genes, including *PAK2*, *RASSF9*, *ITIH1*, *ASH1L*, *SND1*, *AHNAK*, and *MINK1*, differentially expressed in ASD vs. FXSA, were among those listed in the SFARI Gene database. In addition, the *FMR1* gene, as expected, was found differentially expressed in both the comparisons between TD vs. FXSA and ASD vs. FXSA groups.

### 3.5. Functional Implications of Differentially Methylated Sites

The *FMR1* gene showed strong hypermethylation in the FXSA samples, consistent with the epigenetic silencing observed in FXS. GO (gene ontology) and KEGG enrichment pathways analyses were carried out for the 53, 60, and 545 DMS (with high effect size). Surprisingly, there were no significant GO terms and KEGG pathways found for ASD-related DMS. However, we observed that there were some GO terms that were significantly enriched with DMS in the FXSA group, including the regulation of cellular protein catabolic process, regulation of synaptic plasticity, synaptic vesicle exocytosis and regulation of synaptic vesicle cycle.

### 3.6. Validation of Gene Expression of a Subset of Differentially Methylated Genes

Gene expression was measured in 14 selected genes that had a significant differential methylation profile among the 3 groups. Of those, *ZNf587*, *NF2*, and *C11orf31* genes had an RNA expression profile in line with their methylation status. The mRNA expression of *ZNf587* was significantly higher in children with ASD (*n* = 16, *p* = 0.027) and FXS-ASD (*n* = 16, *p* = 0.054) compared to TD (*n* = 8) children, who confirmed the observed hypomethylation profile of *ZNf587* in both children with ASD and FXS-ASD compared to TD. Further, the expression of *NF2* mRNA was significantly lower in participants with ASD (*n* = 16) than the expression in TD children (*n* = 8, *p* = 0.045) and children with FXS-ASD (*n* = 16, *p* = 0.012), which also agreed with the hypermethylation status of CpG sites within the gene observed in children with ASD compared to TD. Finally, the expression of *C11orf31* was significantly higher in FXS-ASD subjects (*n* = 16) compared to TD (*n* = 8, *p* = 0.04) or ASD (*n* = 16, *p* = 0.02), which also agreed with the CpG hypomethylated status observed in children with FXS-ASD compared to ASD and TD subjects. We further determined the plasma amount of the C11orf31 and NF2 proteins in the three groups, ASD (*n* = 23), FXS-ASD (*n* = 21), and TD (*n* = 10), using an ELISA approach. Although there was a trend of higher expression of C11orf31 protein in FXS-ASD compared to TD and ASD, in line with the corresponding RNA expression profile, the difference did not reach the statistical significance. However, NF2 protein was significantly less expressed in ASD compared to both FXS-ASD (*p* = 0.055) and TD (*p* = 0.036), consistent with its lower mRNA expression levels and with the hypermethylated DNA profile.

## 4. Discussion

In recent years, research suggests that complex neurodevelopmental disorders, such as ASD, due to their high heritability, necessitate the involvement of multifactorial causes, likely an interplay between both genetic and environmental factors. Epigenetics links genetic and environmental influences, contributing to the alteration of neurodevelopmental processes [17,48,49].

Hence, researchers studying ASD and FXS have started delving deeper into the possible epigenetic mechanisms that might contribute to the clinical phenotypes characterizing children with these disorders. Usually, studies to identify disease-associated epigenetic markers aim at determining the role of an identified biomarkers as diagnostic factors, descriptors, or modulators for the risk and prognosis of the disorder in patients and to elucidate their roles in the pathogenesis of the disorder [50]. In this study, we used an epigenetics approach to obtain a global methylation profile in three groups of children, ASD, FXSA, and TD, matched by age (2–6 years) and gender.

The comparison between the TD group and the ASD group identified an abnormal methylation pattern of a number of genes, including two genes, *PAK2* and the *FANCD2,* that were significantly differentially methylated. The *PAK2* gene encodes for a serine/threonine protein kinase; it is highly expressed in the fetal brain, plays a role in a variety of different signaling pathways, it is activated by the Rho family GTPases, *Rac*, and *Cdc42*, and it is a regulator of the actin cytoskeleton remodeling dynamic [51]. *PAK2* is important for brain development, and its haploinsufficiency leads to autism-related behavior [7,52] and its inhibition partially restores several synaptic fragile X syndrome phenotypes in the Fmr1 KO mice [53]. Our findings of haploinsufficiency (hypermethylation) of *PAK2* in ASD compared to TD and FXSA, is consistent with the above reports. This suggests a potential cause leading to the observed defects in spine morphogenesis and altered synaptic function and could have significant therapeutic implications for this disease.

*FANCD2* is a component of the Fanconi anemia (FA) DNA repair pathway, and its nuclease activity has been shown to have a protective role against expansion in an HD cell model [54]. Interestingly, the interaction between activated *FANCD2* and *DNMT1* has been demonstrated, indicating that, perhaps, the recruitment of *DNMT1* by *FANCD2* could lead to the methylation modification of genomic DNA [54].

The comparison between the TD group and the FXS group identified three genes in the SFARI database that were differentially methylated. The *GTF2I* gene, a transcription factor *TFII-I* highly expressed in the brain, plays a role in brain development, and its haploinsufficiency has been implicated in the etiology of language-related disorders [55,56,57]. This transcription factor, when activated [58], can enhance or suppress *DYX1C1,* a transcription factor, which binds to nuclear *ESR1* and *ESR2*, and promotes their proteasomal degradation, negatively regulating, therefore, their function [59]. Interestingly, *ESR1* and *ESR2*, as well as the *FMR1* gene, are involved in activating *ERK1/2* [59,60], and elevated *ERK* signaling has been reported in several studies and in brain tissue derived from both human and *Fmr1* knockout mice [61]. *ERK* activation through downstream signaling guides to changes in the cytoskeletal organization, which ultimately leads to the stimulation of neurite outgrowth [62,63]. The *FOXP1* gene, which was hypomethylated in FXSA, compared to TD, encodes for a transcription factor important for early brain development and, interestingly, variants, deletions, missense mutations *FOXP1* are causative for severe forms of ASD, often comorbid with intellectual disability, language deficits, and congenital anomalies, including mild dysmorphic features, and brain, cardiac, and urogenital abnormalities [64].

Among the seven genes differentially expressed in the comparison between ASD vs. FXSA, there were *PAK2*, described above, differentially methylated in ASD vs. TD, and *RASSF9*, a member of the Ras-association domain family (RASSF), whose biological and physiological role is currently unknown. It is expressed in multiple organs, including the testis and brain, and is associated with the recycling of endosomes [65]. Hypermethylation of this gene was observed in the FXSA group compared to the ASD group. In recent studies, the *AHNAK* gene, a multifunctional protein in the brain, was found to be one of the neurodevelopmental disorder risk genes [66], potentially modulating depressive behavior [67].

The clinical relevance of our and others’ findings regarding these genes as part of pathways that are altered in these diseases should be further investigated, as a better understanding of the molecular complexity will contribute to potential novel intervention sites for the development of novel molecular targets for therapeutics in FXS and autism.

Finally, it is worth noting that we carried out a validation of a subset of differentially methylated genes that confirmed that two genes, *NF2* and *C11orf31*, had an RNA expression profile significantly in line with their methylation status. Indeed, significantly decreased expression levels of *NF2* mRNA were detected in the ASD group (hypermethylation was detected at the RNA level) compared to the TD and the FXSA group, while significantly increased expression levels of *C11orfF31* mRNA were detected in the FXSA group (hypomethylation was measured at the RNA level), compared to TD and ASD. Further, expression levels of the proteins encoded by these two genes were found to be consistent with the transcription data, although only the *NF2* reached significance.

The *NF2* gene is involved in the regulation of neurogenesis and neuronal projection development, and is considered, therefore, one of the key contributors to brain development. It maintains a definite balance between the production of post-mitotic neurons and glia cells and the expansion of the neural progenitor pool. The absence of *NF2* leads to the expansion of neural progenitor cells of the hippocampal primordium and other brain regions, resulting in severe malformation of the hippocampus in the *NF2* mutant mice model [68]. The hypermethylation of *NF2* and consequent decreased mRNA and protein expression observed in the ASD group in our study may contribute to the reduced hippocampal connectivity and structural and synaptic deficits in hippocampal regions observed in children with ASD.

*C11orf31* encodes for selenoprotein-H which is known for its antioxidant function in the brain [8]. The increased *C11orf31* expression in FXSA might be attributed to a feedback neuroprotective mechanism trying to counteract the oxidative stress documented in FXSA [69]. Additionally, selenoprotein-H was found to play a role in tumorigenesis prevention [70], and the increased expression we observed in FXSA could also represent one of the protective mechanisms against cancer proposed in individuals with FXS [71].

This study has several limitations. First, the sample size warrants confirmatory study in a larger cohort. Second, although we presented our findings taking into consideration blood cell heterogeneity, a DM profile could be tissue specific and could have the potential of confounding DNA methylation measurements. Third, intergenic regions are known to have a significant regulatory impact on the expression of various genes. Since 40–60% of DMs in the three-comparison group are located within the intergenic regions, our preliminary observations from this study will require further investigations to understand the impact and the role of DM profiles in the pathogenesis of ASD and FXS.

Overall, in this study, we used an epigenetics approach to obtain a global methylation profile in three groups of children, ASD, FXSA and TD, matched by age (2–6 years) and gender. We characterized the methylome profile in the three groups which distinguishes them, suggesting a potential role for altered DNA methylation in the pathology of these disorders. Additionally, these findings may help in the diagnostic classification of children with ASD and FXSA and may pave the way for developing novel treatment modalities that could reverse the altered methylome profile in children with neurodevelopmental disorders.

## 5. Conclusions

This study profiled methylome variations in ASD and FXSA and showed a potential role for altered DNA methylation in the pathology of these disorders, suggesting that epigenetic mechanisms may mediate some components of the disease during early neurodevelopment. These findings may help in the diagnostic classification of children with ASD and FXSA and may pave the way for developing novel treatments.

## Figures and Tables

**Figure 1 genes-13-01795-f001:**
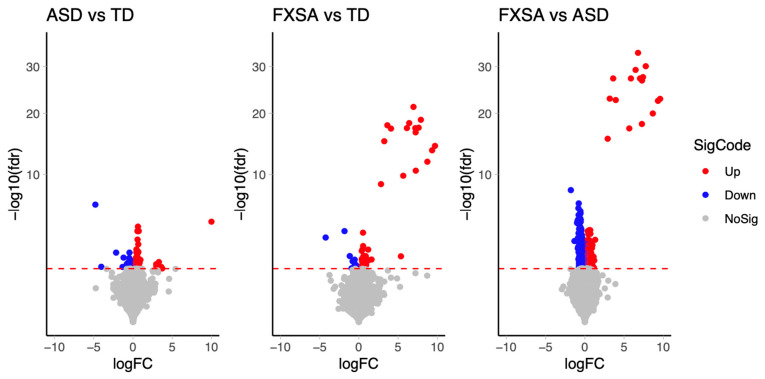
Volcano plot of DMS. Volcano plots for the significant DMS that were identified with adjusted *p*-value < 0.05 in the three groups comparison, ASD vs. TD, FXSA vs. TD and FXSA vs. ASD, where blue circles indicate significant hypomethylation probes with adj *p*-value < 0.05 and red circles indicate hypermethylation probes with adj *p*-value < 0.05.

**Figure 2 genes-13-01795-f002:**
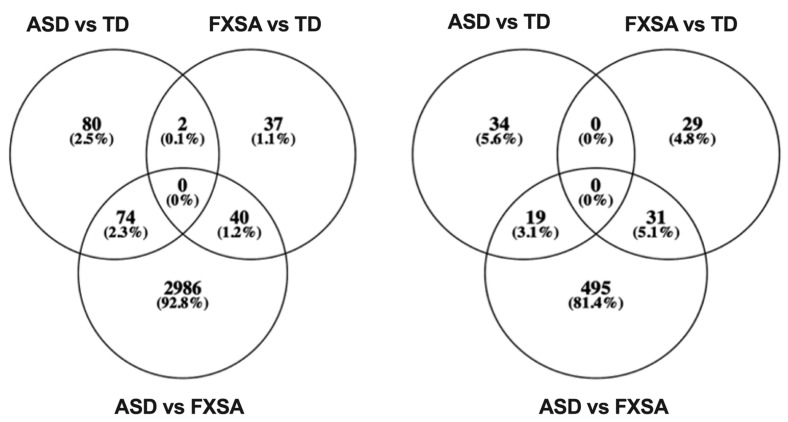
Venn diagram comparison of DMS among the different disease groups. The left Venn diagram shows the DMS, which were identified by FDR ≤ 0.05. On the right, the Venn diagram shows the DMS, identified by FDR ≤ 0.05 and absolute difference of beta value ≥ 0.05.

**Table 1 genes-13-01795-t001:** Demographic and molecular characteristics of participants.

	*TD*	*ASD*	*FXSA*
	(n = 11)	(n = 23)	(n = 23)
** *Age (Years)* **			
Mean (SD)	3.8 (1.3)	4.1 (1.1)	3.6 (1.1)
Median (Range)	4 (2–6)	4 (2–6)	4 (2–6)
** *CGG Repeats* **			
Mean (SD)	28.5 (3.1)	28.7 (3.4)	(all > 200)
Median (Range)	30 (21–31)	30 (21–33)	
** *Mutation Category* **			
Full	0	0	12 (52.2%)
Mosaic (Size/Methylation)	0	0	11 (47.8%)
No Mutation	11 (100%)	23 (100%)	0

**Table 2 genes-13-01795-t002:** Number of Differentially Methylated Sites (DMS) identified by the adjusted model.

	*TD vs. ASD*	*TD vs. FXSA*	*ASD vs. FXSA*
FDR ≤ 0.05	156	79	3100
FDR ≤ 0.05 & BetaAbsDiff ≥ 0.05	53	60	545
FDR ≤ 0.05 & BetaAbsDiff ≥ 0.1	14	38	42

## Data Availability

Data and results generated from this project will be fully available upon request. Biological samples from subjects included in this study will be available under MTA agreement accordingly to the University of California, Davis policy.

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
