# Peer review of "Profiling Genome-Wide DNA Methylation in Children with Autism Spectrum Disorder and in Children with Fragile X Syndrome"

_genes, 2022, doi:10.3390/genes13101795_

Round 1
Reviewer 1 Report
Dear Author,
Thanks for submitting your research manuscript entitled " Global Methylation Profiling in Children with Autism Spectrum Disorder and in Children with Fragile X Syndrome ".
Before giving my final comments as well as the final revision of this manuscript, firstly, author needs to address the following comments scientifically.
Major concerns: Please find out the following comments
The rationale and purpose behind selecting the
· Fragile X syndrome (FXS) in ASD mental health system is incomplete.
· Title is misleading the reader. Title needs to reframed in simply manner accordingly.
· The reviewer found irrational and non-scientific justification in the abstract—introduction and discussion part.
· Abstract is very poorly written and very confusing. Irrational and fused with repetitions. Scientific output is not clear with this abstract.
· The results and discussion are very poorly explained. Reviewer surprise to see the justification of conclusion part
· The reviewer feels the author needs to elaborate and justify it with proper citations and strong evidence. The author fails to explain the relevant justification in the introduction as mentioned in the discussion part.
· A major drawback is a lack of supporting pre-clinical and clinical evidences.
· Complete mismatch of abstract, introduction, results and discussion in concern with ASD.
Title:
· Mismatch of title with relevant introduction and conclusive remarks in the conclusion part.
Abstract:
- The rationale behind this research is not well explained, and several major concerns still constrain the reviewer's enthusiasm for publishing this manuscript. And not suitable for readers.
Example 1:
In this study, we determined the global DNA methylation profiles of biological samples de-rived from 57 age-matched male participants (2-6 years old) including 23 subjects with ASD, 23 subjects with FXS with ASD (FXSA) and 11 typical developing (TD) children. Human Methylation EPIC Bead Chips, including 850,000 CpG sites throughout the genome, were used for this study.
Example 2:
Human Methylation EPIC Bead Chips, including 850,000 CpG sites throughout the genome, were used for this study. After controlling for technical variation and white blood cell composition, using conservatory threshold of false discovery rate (FDR<=0.05) we identified 156, 79 and 3100 differentially methyl-ated sites (DMS) in TD vs AD, TD vs FXSA and ASD vs FXSA comparison groups, respectively (FDR<0.05). Of these, 53, 60 and 545 DMS in TD vs AD, TD vs FXSA and ASD vs FXSA comparison groups has absolute beta-value difference of at least 0.05.
Introduction:
- The basic literature is not well written and does not even include any literature on alternative approaches with updated references regarding involvement of current drug treatment/techniques
- Authors fail to justify the correlation, and almost irrational and common information is present in the introduction part.
Material and methods:
- Major drawback is the lack of supporting references and incomplete experimental paradigms.
- Provide biochemicals kits numbers along with their city, country in all individual parameters in all etc.
- Lack of ethical guidelines and approval number is major concern.
- In order to support the assessment of all mentioned parameters in his study, the author should provide all the source documents and data he/she has followed for all assays and estimates.
- How was the sample size determined? Ideally, a priori sample size calculation should be performed to determine the appropriate sample size.
- Normality and variance homogeneity should be assessed across all groups of the same outcome variable and not individual experimental groups. If the data were not normally distributed or variance homogeneity was not met, nonparametric tests need to be performed. Parametric data should be reported as mean +/- SD, while nonparametric data should be given/displayed as median and interquartile range. Longitudinal data should be analyzed using repeated measures tests.
Results:
- Fir better understanding, and reader, divide and convert all tables 1 and 2 into figures. Tables are not accepted in current form.
- Results need more clarification and significant justification. Differentiating between the outcome and the discussion sections is quite difficult.
- Use proper statistical reporting: i.e. for the results of each statistical test, the authors should report the statistical test that was applied, the test statistic (e.g. t, U, F, r), degrees of freedom as subscripts to the test statistic, and the exact probability value, including those for normality and variance homogeneity tests. Statistics should be reported in APA format, i.e.: t(df) = value, p = value; F(df1,df2) = value, p = value; r(df) = value, p = value; [chi]2 (df, N = value) = value, p = value; Z = value, p = value.
- Include statements on the tests for normality and variance heterogeneity and respective results. If the data were not normally distributed or variance heterogeneity was not met, nonparametric tests need to be applied.
Discussion:
- To address the outcome of in-vivo measures/results separately and how they correlate with the existing literature, it would be better if the author restructured to take a more critical approach.
- In the discussion and the conclusion, the aims, rationale, and future perspectives are not evident clearly in relation with in-vitro and in-vivo experimentation.
- Add the limitations at the end of the discussion part.
- The discussion is usually unorganized at the beginning to address all the observations and evaluate them at the end. It makes the results easier to contextualize and simpler to comprehend.
- Furthermore, a minimal critical analysis should be provided, along with current study limitations as well the future perspective as separate paragraph.
Conclusion:
- Need to revise the conclusion in a scientific manner. Not accepted in its current form.
Example : This study profiled methylome variations in ASD and FXSA and showed a potential role for altered DNA methylation in the pathology of these disorders suggesting that ep-igenetic mechanisms may mediate some components of the disease during early neuro-development. These findings may help in the diagnostic classification of children with ASD and FXSA and may pave the way for developing novel treatments.
- This reviewer considers that this paper cannot be published in the present form. A detailed revision shortening, ordering and following the commented ideas could improve this interesting paper in a significant manner.
- Several typewriting mistakes are present and needing correction. This reviewer remains at entire disposal for the next version.
Reviewer 2 Report
- this study answers the question about the Global DNA methylation profile in the pathology of ASD and fragile x.
- This study will add a new perspective for developing therapeutic interventions that could reverse the altered methylome profile in children with neurodevelopment disorder
Author Response
We thank the reviewer for expressing the importance of our study
Round 2
Reviewer 1 Report
Dear Author,
After careful consideration, Manuscript can be accepted for publication.